# Influence of Bacterial Secondary Symbionts in *Sitobion avenae* on Its Survival Fitness against Entomopathogenic Fungi, *Beauveria bassiana* and *Metarhizium brunneum*

**DOI:** 10.3390/insects13111037

**Published:** 2022-11-09

**Authors:** Sajjad Ali, Asif Sajjad, Qaiser Shakeel, M. Aslam Farooqi, M. Anjum Aqueel, Kaleem Tariq, Muhammad Irfan Ullah, Aamir Iqbal, Aftab Jamal, Muhammad Farhan Saeed, Barbara Manachini

**Affiliations:** 1Department of Entomology, The Islamia University of Bahawalpur, Bahawalpur 63100, Pakistan; 2Department of Plant Pathology, The Islamia University of Bahawalpur, Bahawalpur 63100, Pakistan; 3Department of Agriculture Entomology, Abdul Wali Khan University, Mardan 23200, Pakistan; 4Department of Entomology, University of Sargodha, Sargodha 40100, Pakistan; 5Department of Crop Sciences, Georg-August University, 37073 Goettingen, Germany; 6Department of Soil and Environmental Sciences, Faculty of Crop Production Sciences, The University of Agriculture, Peshawar 25130, Pakistan; 7Department of Environmental Sciences, COMSATS University Islamabad, Vehari Campus, Vehari 61100, Pakistan; 8Department of Agricultural, Food and Forest Sciences (SAAF), University of Palermo, Viale delle Scienze 13, 90128 Palermo, Italy

**Keywords:** entomopathogenic fungi, bacterial symbiont, *Hamiltonella defensa*, *Regiella insecticola* wheat aphid, biological control, *Beauveria bassiana*, *Metarhizium brunneum*

## Abstract

**Simple Summary:**

Symbiotic relationships are common in nature and have influenced the evolution of life across the cosmos. Bacterial secondary symbionts are linked to the evolution of aphids’ heightened natural enemy defense mechanisms. Due to their significant historical contributions to insect control, entomopathogenic fungi are recognized as potential biocontrol agents. The research highlights the role of symbiotic bacteria in the protection of insects, especially aphids, against the entomopathogenic fungi and discuss potential implications of this symbiosis.

**Abstract:**

The research was focused on the ability of wheat aphids *Sitobion avenae*, harboring bacterial secondary symbionts (BSS) *Hamiltonella defensa* or *Regiella insecticola*, to withstand exposure to fungal isolates of *Beauveria bassiana* and *Metarhizium brunneum*. In comparison to aphids lacking bacterial secondary symbionts, BSS considerably increased the lifespan of wheat aphids exposed to *B. bassiana* strains (Bb1022, EABb04/01-Tip) and *M. brunneum* strains (ART 2825 and BIPESCO 5) and also reduced the aphids’ mortality. The wheat aphid clones lacking bacterial secondary symbionts were shown to be particularly vulnerable to *M. brunneum* strain BIPESCO 5. As opposed to wheat aphids carrying bacterial symbionts, fungal pathogens infected the wheat aphids lacking *H. defensa* and *R. insecticola* more quickly. When treated with fungal pathogens, bacterial endosymbionts had a favorable effect on the fecundity of their host aphids compared to the aphids lacking these symbionts, but there was no change in fungal sporulation on the deceased aphids. By defending their insect hosts against natural enemies, BSS increase the population of their host society and may have a significant impact on the development of their hosts.

## 1. Introduction

Symbiosis is a close, long-term, mutual relationship between two organisms of different species that may or may not benefit each other [1]. Symbiotic associations are ubiquitous in nature and have played a vital role in the evolution of life on Earth [2]. Among the diverse examples of symbiotic associations, the most sophisticated ones have been reported to be associated with bacterial endosymbionts, which allow for intimate interactions in the partners involved [3]. Insects are the most diverse order of animals and almost half of all insect species are predicted to harbor bacterial endosymbionts [4]. In most cases, bacterial endosymbionts are vertically transmitted from one generation to the next generation [5]. Bacterial endosymbionts in insects are categorized into two groups: (1) obligate or primary symbionts and (2) facultative or secondary symbionts. Primary symbionts are obligate in nature for host survival. They reside in specialized cells in their hosts, called bacteriocytes, and are mutually beneficial for both partners [6]. *Buchnera aphidicola* Munson (Enterobacterales: Erwiniaceae) in aphids and *Candidatus Portiera aleyrodidarum* Thao and Baumann (Oceanospirillales: Halomonadaceae) in whiteflies are typical examples [5,7,8,9]. *B. aphidicola* and their aphid associations display an outstanding relationship among bacteria and insects, where none of the partners can survive without the other. The bacterium contributes more than 90% of the essential amino acids needed by the pea aphid, *Acyrthosiphon pisum* Harris (Hemiptera: Aphididae), and directly contributes to nutritional fitness [10].

Bacterial secondary symbionts (BSS) facultatively colonize their hosts, occurring more occasionally and found living freely in the hemolymph or other tissues of their hosts. They have the ability to influence the developmental, morphological, and physiological traits of their hosting individuals, which may benefit their spread and further establishment within host populations [11]. The best-studied BSS–aphid interactions used the pea aphid *A. pisum*–symbiont system. In this species, BSS contributed to the thermal fitness of aphids against heat shock by improving their survival and fecundity [12,13]. They have also been reported to contribute to the effective utilization of host plants [14,15], and it is hypothesized that geographic variation in host plant use is also influenced by BSS [16]. BSS also contribute to the protection of aphids against natural enemies [17]. Moreover, pea aphids harboring *Regiella insecticola* Moran (Enterobacterales: Enterobacteriaceae) have been reported to exhibit increased resistance against infection by the entomopathogenic fungus *Pandora neoaphidis* Remaud and Hennebert (Entomophthorales: Entomophthoraceae) [18]. Similarly, *R. insecticola*, *Rickettsia* sp., and *Rickettsiella* sp. reduced the mortality of their host aphids and also decreased the sporulation of fungus on dead aphids [19]. *R. insecticola* provided resistance against the aphid-specific fungal pathogen *Zoophthora occidentalis* (Thaxt.) A. Batko (Entomophthorales: Entomophthoraceae), while no resistance against a generalist fungal entomopathogen such as *Beauveria bassiana* (Bals.-Criv.) Vuill. (Hypocreales: Cordycipitaceae) was reported [20]. The BSS *Hamiltonella defensa* Moran (Entomophthorales: Entomophthoraceae) provided protection to pea aphids against the parasitoid *Aphidius ervi* Haliday (Hymenoptera: Braconidae) by killing its larvae in the aphids [15,21]. *H. defensa* and *Serratia symbiotica* Moran (Enterobacterales: Yersiniaceae) in *A. pisum* also reduced consumption by a predatory ladybird beetle [22]. There is now sufficient evidence that BSS have a prominent influence on the ecology and evolution of their hosts [23] and are also impacting the population dynamics of their hosts [24,25]. By providing protection to their hosts, secondary endosymbionts increase in frequency within host communities [19].

The English grain aphid *S. avenae* is an economically important insect pest in Western and Central Europe, causing severe damage via quantitative and qualitative losses in wheat crops [26,27,28,29]. Traditionally, insecticides are one of the most common control measures used against aphids, as they are cheap and fast-acting; however, aphids have the ability to rapidly develop insecticide resistance and to reproduce very quickly, making insecticidal treatments less efficient [30,31]. In addition, insecticides are found almost everywhere, and this contamination places the environment and non-target organisms, ranging from beneficial soil microorganisms to insects, fishes, and birds, at increased risk [32].

Thus, there is a need to develop environment friendly and cost-effective measures to reduce pest aphid populations. Biological control using entomopathogenic fungi (EPF) of wheat aphids is a good option to replace chemical control [33]. As compared to the conventional control by pesticides, EPF are environmentally friendly, with no or less risk of leaving pesticide residues in food, and are beneficial for the conservation of the biodiversity of natural enemies in agroecosystems [34]. The mode of action of entomopathogenic fungi involves the germination and penetration of infective spores into the cuticle, followed by developing a germ tube in the hemocoel of the insect host. Thereafter, the fungus reproduces in the insect and causes death due to the production of toxins and/or multiplication to inhabit the entire insect [35].

Approximately 700 fungal species have been described for their entomopathogenic effects, with *Beauveriabassiana* and *Metarhizium brunneum* Petch (formerly *Metarhizium anisopliae* var. anisopliae) (Hypocreales: Clavicipitaceae) being the best-studied species [35,36,37]. *B. bassiana* is a generalist entomopathogen due to possessing a stereotypical pattern of pathogenicity genes towards many insect species [38].

Several myco-pesticides, using EPF species such as *B. bassiana*, *M. anisopliae*, and *Akanthomyces lecanii (Zimm.)* Spatafora, Kepler & B. Shrestha (Hypocreales: Cordycipitaceae), have been registered for the control of aphids [39,40]. However, biological control of insect pests by these fungal pathogens on a commercial level needs to take into account the diverse ecological interactions between, e.g., the pest species and the entomopathogens on a species and strain level [41].

Although several studies have already addressed the role of BSS in aphid life history parameters, to the best of our knowledge, there are no studies systematically evaluating the aphid genotype–BSS interactions with regard to their role when exposed to different fungal strains. Here, we used two genotypes of *S. avenae* either harboring or lacking BSS to assess the role of these BSS in conferring resistance against EPF isolates of *B. bassiana* (strains EABb04/01-Tip and Bb 1022) and *M. brunneum* (strains ART2825 and BIPESCO 5). We evaluated the role of these BSS in survival, fecundity, and mortality in wheat aphid over time after exposure to EPF.

## 2. Materials and Methods

### 2.1. Insect Cultures

Two clones of the English grain aphid, *S. avenae*, named 5 and 7, were used in this study. These clones were initially collected near Giessen (Giessen, Germany). These clones originally harbored the bacterial secondary symbionts *H. defensa* and *R. insecticola* and were reproduced from a single parthenogenetic female aphid [42]. Aphids were maintained under laboratory conditions on winter wheat plants (cv Dekan, KWS GmbH., Einbeck, Germany).

### 2.2. Exclusion of BSS

Antibiotics have been proven to be a tool for the removal of facultative symbionts from their natural hosts [16,43]. We used an antibiotic micro-injection protocol for the removal of *H. defensa* from aphid clone 5 [44]. The injection dose contained a mixture of 250 µg of ampicillin, cefotaxime, and gentamycin per ml solution. For the removal of *R. insecticola* from aphid clone 7, ampicillin was injected with a dose of 1 µg/mg of aphid body weight [45]. Second-instar aphids were paralyzed by exposing them to CO_2_ for 20 s before injecting the antibiotics. Aphids treated with antibiotics were separately transferred to wheat plants and permitted to reproduce. The nymphs produced from treated aphids were named generation 1 (G1). From these G1 nymphs, 4–5 nymphs were collected randomly from each antibiotic-treated mother aphid and raised on wheat plants until they were able to reproduce sufficient progeny as generation 2 (G2). To confirm the absence of bacterial symbionts, G1 aphids were subjected to diagnostic PCRs. Then, only those G2 nymphs were allowed to proceed to further rearing whose G1 mothers displayed the successful elimination of bacterial secondary symbionts (BSS). Diagnostic PCR was performed up to the eighth generation before trials started, to confirm the exclusion of BSS from aphids. Finally, four aphid lines, named clone +5, clone −5, clone +7, and clone −7, respectively, were prepared. Aphid clones assigned (+) harbored BSS, while clones assigned (–) lacked BSS, based on an identical genetic background.

### 2.3. PCR Protocol for the Detection of BSS

DNA extraction from aphid clones was carried out by following the CTAB procedure [46]. Diagnostic PCR, by using specific primers, was performed to detect the presence or absence of BSS by amplifying 16S rDNA gene fragments. Forward primer HDFn [5-ATGAAGTCGCGAGACCAAA-3] and reverse primer HDRn [5-GCTTTCCCTCGCAGGTTC-3] were used for *H. defensa*, and forward primer RIFn [5-GAAGGCGGTAAGAGTAATATGC-3] and reverse primer RIRn [5-CCCCGAAGGTTAAGCTACCTA-3] were used for *R. insecticola* detection, respectively. The following temperature profile was used for the diagnostic PCR: 94 °C for 3 min followed by 30 cycles of 94 °C for 30 s; 60 °C for 40 s; 72 °C for 90 s, and concluding incubation was performed at 72 °C for 8 min. The PCR was performed in a 25 µL volume with one µL of the DNA template having 0.32 µM of each primer, 2 mM MgCl_2_, 200 µM dNTP’s, 1× “Bioline” PCR buffer, and 0.25 units of Taq DNA polymerase. PCR products were observed by using ethidium bromide on 1.7% agarose gel. For the verification of BSS, the PCR products were purified from the gel and sent for sequencing to LGC Genomics GmbH, Germany. The final sequences were then correlated with the known sequences of *H. defensa* and *R. insecticola*, utilizing the BLAST algorithms at NCBI to confirm the identity of the endosymbionts.

### 2.4. Fungal Cultures

Fungal isolates (Table 1) were cultured on potato dextrose agar (PDA) (Sigma-Aldrich, Merck KGaA, Darmstadt, Germany) medium in 90 mm Petri dishes. PDA media were prepared by mixing 39 g PDA mixture (potato extract: 4 g, dextrose: 20 g, agar: 15 g) in 1000 mL Bidest water and autoclaving them for 20 min at 121 °C. Petri plates with PDA mixtures were inoculated with fungal spores and incubated for 2 weeks at 20 ± 1 °C with a 16:8 (L:D) hour photoperiod for mass spore production to be used for bioassays.

### 2.5. Bioassay

The roots of young wheat plants were wrapped with moist cotton and placed in 90 mm Petri dishes with filter sheets. In sterile clean bench circumstances, 10 *S. avenae* nymphs (second or third instar) were transferred to these Petri plates. Using the approach in [51], spore suspensions from entomopathogenic fungi were created after harvesting the spores from PDA plates. In sterilized conical flasks, harvested conidial spores were combined with 100 mL of 0.03% tween TM 20–sterile water solution, and the suspension was agitated using a magnetic stirrer (Bioevopeak Co., Ltd., Jinan, China). By counting the spores in a hemocytometer (Sigma-Aldrich, Merck KGaA, Darmstadt, Germany) while they were being observed under a microscope (Bioevopeak Co., Ltd., Jinan, China the final concentrations of the spore suspensions were determined and corrected for 1 × 10^8^ spores per ml. To apply spores to aphids on Petri plates, the fungal spore suspensions were placed into a hand sprayer (Bürkle GmbH, Bad Bellingen, Germany). Each plate received a complete shower of a spore suspension sprayed from a certain distance (8–10 inches) to cover the whole plate area. All of the plates were para-film-sealed after spore applications, and they were all placed in a climate-controlled room with a 16:8 (L:D) hour photoperiod at 20 *±* 1 °C and 65 ± 5% relative humidity. The aphid clones +5, −5, +7, and −7, with 16 replications each, were sprayed using an identical technique for all entomopathogenic fungal strains. The aphids in the control group were sprayed with a 0.03% tween TM 20 solution diluted in 100 mL of sterilized water. For up to five days, dead aphids were counted daily to track the progression of the fungus infection and determine the overall mortality.

### 2.6. Fecundity Assay

The indirect effects of fungal strains on the fertility of aphids harboring or not harboring bacterial secondary symbionts were evaluated using a fecundity experiment. After recording the final mortality on day five, 16 wingless adult aphids were randomly chosen, and they recovered from the entomopathogenic fungal treatment. These aphids were housed in climate-controlled rooms alongside wheat plants in fresh Petri plates. Daily fecundity data were collected for five days after the females from the first neonate. For a maximum of 16 days, the total number of offspring generated was recorded. Data on fecundity were also gathered for the control treatments in a similar manner.

### 2.7. Fungal Growth Assay

The effect of bacterial secondary symbionts on fungal development was evaluated in the dead aphids on a semi-selective medium (SM). First, we produced Sabouraud dextrose medium by dissolving glucose and peptone in demineralized water and bringing the mixture’s pH to 6.3 using 1M HCl. The medium included 10 g of peptone, 20 g of glucose, and 18 g of agar per liter of solution. Agar was then added when the solution was transferred to a volumetric flask. The media were autoclaved at 121 °C for 20 min, before being cooled to 50 °C. Before transferring the media to Petri dishes, antibiotics (Thermo Fisher Scientific, Waltham, MA, USA) (cycloheximide: 50 mg, streptomycin: 100 mg, tetracycline: 50 mg per liter) and fungicides (dodine: 100 mg per liter) were applied [52]. On this SM medium, dead aphids from the aforementioned treatments were applied (5 aphids per plate). At 5, 7, and 9 days after inoculation, the diameters of the developing fungi from dead wheat aphids were measured in millimeters using a vernier scale.

### 2.8. Statistical Analysis

In a one-way ANOVA, the total aphid mortality was analyzed using the least significant difference (LSD) test. Using repeated-measure analysis of variance (ANOVA) and a paired t-test, analyses of the fecundity of aphids and fungal radial growth after fungal treatments were performed. Statistix (version 8.1) software was used to conduct statistical analyses, with a 5% threshold of significance [53]. The Kaplan–Meier survival tool in R software (version 3.1) was used to calculate the daily aphid survivability rate.

## 3. Results

### 3.1. Total Mortality in Wheat Aphid

The strains of entomopathogenic fungi, *B. bassiana* and *M. brunneum*, varied with regard to pathogenicity and virulence against aphid clones with and without BSS, causing significantly higher mortality in those clones without bacterial symbionts. *B. bassiana* strains Bb1022 and EABb04/01-Tip and *M. brunneum* strains ART 2825 and BIPESCO 5 caused 70, 59, 46, and 80% mortality, respectively, in clone –5 after five days, while 30, 19, 16, and 31% mortality, respectively, was recorded in clone +5 (Figure 1a). The bacterial secondary symbiont *H. defensa* partially protected aphids harboring them, significantly reducing mortality (*p* < 0.005) as compared aphids lacking *H. defensa*. Similar results in mortality were recorded for clone −7 when exposed to *B. bassiana* strains Bb1022 and EABb04/01-Tip and *M. brunneum* strains ART 2825 and BIPESCO 5, causing 80, 56, 61, and 88% mortality, respectively, after 5 days, while 35, 28, 21, and 63%, respectively, was recorded for clone +7 (Figure 1b). *M. brunneum* strain BIPESCO 5 was found the most lethal to aphids without BSS as compared to the other fungal strains tested and caused more than 80% mortality to clone −5 and −7. The mortality of wheat aphids in control treatments was less than one percent.

### 3.2. Fungal Radial Growth

Both tested entomopathogenic fungi, *B. bassiana* and *M. brunneum*, successfully sporulated on dead aphids placed on semi-selective media. The radial growth of *B. bassiana* strain Bb1022 on dead aphids either harboring or not harboring *H. defensa* or *R. insecticola* was not significantly different (*p* > 0.005) after 5, 7, and 9 days of inoculation and showed an almost similar growth pattern. Similar results were obtained in the case of *B. bassiana* strain EABb04/01-Tip for both of the aphid clones. Once EPF were able to overcome the resistance conferred by the bacterial secondary symbionts, due to the death of the specimens colonized, no difference in fungal growth on dead aphids with or without bacterial secondary symbionts was observed. Meanwhile, the isolate of *M. brunneum* ART 2825 showed significantly less radial growth on aphids harboring *H. defensa* as compared to those lacking symbiotic bacteria up to seven days after plating them on agar (*p* < 0.005); however, at day nine, this difference was no longer significant (*p* > 0.005). The same strain, ART 2825, behaved differently on wheat aphid clones 7 and −7. Differences in radial growth from aphids with and without *R. insecticola* were non-significant from the beginning up to 7 days, but growth significantly accelerated on clone −7 (*p* < 0.005) from aphids containing *R. insecticola*. *M. brunneum* BIPESCO 5 grew equally well (*p* > 0.005) on dead aphids with or without *H. defensa* for up to 9 days (Table 2). In the case of aphids harboring *R. insecticola*, less growth was observed in the case of BIPESCO 5 for the first 5 days after plating dead aphids onto the media, as compared to aphids without *R. insecticola*. Thereafter, fungal growth increased in aphids infected with *R. insecticola* and growth differences were non-significant (*p* > 0.005) when comparing aphids with and without *R. insecticola* after 7 and 9 days (Table 2)

### 3.3. Fecundity Assay

Fecundity in wheat aphids harboring *H. defensa* or not harboring bacterial symbionts was not significantly different (*p* > 0.005) in control treatments and resulted in 10 nymphs per female aphid in five days. Similar results were recorded in aphids harboring or not harboring *R. insecticola*. Female aphids, infected with *H. defensa* and *R. insecticola* exposed to the *B. bassiana* strain Bb1022, exhibited a significantly higher (*p* < 0.005) reduction in cumulative fecundity. Within five days, female aphids harboring *H. defensa* and *R. insecticola* produced an average of 4.62 ± 0.22 and 4.44 ± 0.46 nymphs, respectively, as compared to aphids lacking them (2.56 ± 0.22 and 1.75 ± 0.18 nymphs per female). Similar fecundity effects were observed for *B. bassiana* strain EABb04/01-Tip, either with or without *H. defensa* and *R. insecticola*, producing 4.88 ± 0.42 and 5.25 ± 0.64, or 2.13 ± 0.44 and 1.81 ± 0.18 offspring, respectively. The bacterial symbionts *H. defensa* and *R. insecticola* exhibited higher resistance to *M. brunneum* ART 2825 and resulted in higher fecundity as compared to all other fungal strains; however, differences were still significantly different (*p* > 0.005) when comparing aphid clones +5, −5, +7, and −7, producing 7.37 ± 0.66, 3.69 ± 0.26, 7.69 ± 0.94, and 4.94 ± 0.55, respectively (Figure 2). The effect of BIPESCO 5 on the fecundity of aphids harboring *H. defensa* was highly significant (*p* < 0.005), producing 5.56 ± 0.62 nymphs per female, as compared to wheat aphids without *H. defensa*, producing only 1.75 ± 0.44 nymphs. Similar effects of *M. brunneum* BIPESCO 5 were observed in aphid clones with or without *R. insecticola* (*p* > 0.005).

### 3.4. Infection Period

Daily evaluations of aphid survival up to 5 days after fungal spore treatments and of the controls were conducted. The survival of aphids in the control treatments was unaffected by *H. defensa* and *R. insecticola*, with 100% of them still alive after 5 days. *B. bassiana* strain Bb1022 infection began in aphid clones +5 and +7 after the third and second days, respectively, but in aphid clones −5 and −7, infection began within 24 h, leading to early death. The fungal infection was highest on the 5th day after application and resulted in 72% and 70% survival of aphids in clones +5 and +7, while, in clones −5 and −7, survival was significantly reduced to 36% and 25%, respectively. This clearly indicates that *R. insecticola* confer significant resistance to their hosts, and the difference illustrates that the protection against *B. bassiana* strain Bb1022 provided by *H. defensa* and in survival is significant (LR χ^2^ = 334, 399, df = 3 and *p* < 0.005). *H. defensa* and *R. insecticola* delayed *B. bassiana* strain EABb04/01-Tip infections in individuals of aphid clones +5 and +7 (51 and 53%, respectively) compared to individuals without bacterial symbionts, which was highly significant (LR χ^2^ = 270, 224, df = 3 and *p* < 0.005). *M. brunneum* ART 2825 significantly reduced the survival of aphids in clonal lines −5 and −7 as compared to aphids harboring *H. defensa* and *R. insecticola* (LR χ^2^ = 189, 290, df = 3; *p* < 0.005). The infection started earlier in wheat aphids without bacterial symbionts, reducing the survival on the 5th day to 63 and 51%, as compared to the aphid +5 and +7 specimens with a survival rate of 85 and 83%, respectively (Figure 3). *M. brunneum* BIPESCO 5 proved to be the most virulent strain towards both aphid lines without *H. defensa* and *R. insecticola*, reducing their survival to 30 and 19% on day 5 as compared to aphids harboring bacterial symbionts, with a significantly higher survival rate (LR χ^2^ = 458, 484, df = 3 and *p* < 0.005) of 74 and 48%, respectively.

## 4. Discussion

It is evidenced through our results that bacterial secondary symbionts *H. defensa* and *R. insecticola* can influence their host wheat aphids in multiple ways, such as host survival over the infection period against entomopathogenic fungi *B. bassiana* and *M. brunneum* strains. Our results indicate that the presence of bacterial secondary symbionts delays the infection penetration of entomopathogenic fungi into the wheat aphids harboring *H. defensa* and *R. insecticola* as compared to those lacking them, and increases the probability of host survival. Some previous studies have reported that *R. insecticola* increased the survival of its host pea aphid against fungal entomopathogen *P. neoaphidis*, although this effect differed in magnitude among pea aphid clones [18]. Carrying *R. insecticola* and *Rickettsia* spp. was beneficial to pea aphids and showed strong protection effects against entomopathogenic fungi *P. neoaphidis* by enhancing their survival [19]. Our results also highlight that *M. brunneum* strain BIPESCO 5 is the most virulent against wheat aphids, causing up to 90% mortality.

At present, *M. brunneum* strain BIPESO 5 and strain CB15-III are the only certified *Metarhizium* strains for pest control on commercial levels in many European countries [37].

The comparison of the entomopathogenic fungi *B. bassiana* and *M. brunneum*’s fungal sporulation growth on dead aphids with and without *H. defensa* and *R. insecticola* proved that bacterial secondary symbionts cannot influence the fungal growth on their hosts after death, because no appreciable difference in growth rates was found on aphids with or without these bacterial symbionts. This indicates that when the fungal pathogen effectively overcame the resistance provided by the bacterial secondary symbionts at the beginning of the infection, the wheat aphids sporulated successfully, without any difference regarding the presence or absence of bacterial symbionts.

Some previous studies also reported similar results regarding the entomopathogenic fungi *P. neoaphidis* on dead pea and wheat aphids with or without *H. defense.* In fact, the sporulation of *P. neoaphidis* was not affected by the presence of bacterial symbionts after the death of their hosts [19,54]. It is possible that secondary symbionts are only effective against entomopathogenic fungal spread while in combination with the natural immune system of their host aphid, but they lacked effectiveness after the death of their host.

The tested fungal pathogen strains of *B. bassiana* and *M. brunneum* had negative effects on the fecundity of wheat aphids without *H. defensa* and *R. insecticola*. In our results, this was an indirect effect of the entomopathogenic fungi, which caused higher mortality in wheat aphids without bacterial secondary symbionts after 5 days, resulting in reduced fecundity indirectly. This adverse effect caused by the entomopathogenic fungi was successfully reduced in wheat aphids carrying bacterial secondary symbionts by enhancing their longevity under stress. Such effects of entomopathogenic fungi on the aphid’s fecundity are important when considering them as biocontrol agents against aphids, because a decrease in fecundity after infection must contribute to a reduction in populations of wheat aphids. Negative effects on fecundity caused by *M. anisopliae* and *Aspergillus ochraceus* Wilhelm (Eurotiales: Trichocomaceae) fungi have already been reported in *Ceratitis capitata* (Diptera: Tephritidae) [55], while the application of *P. neoaphidis* reduced the fecundity of pea aphids carrying *Rickettsia* and *Spiroplasma* bacterial symbionts [50]. It is inferred that different bacterial symbionts express themselves differently depending on their host and natural enemy’s genetic backgrounds.

Previous studies demonstrated that the host defense of bacterial secondary symbionts was limited to *P. neoaphidis* mainly in aphids, but there should be more explanatory and diversified effects on their host phenotypes against the natural enemies for their persistence in aphid populations. The effectiveness of bacterial symbionts is dependent upon the genotypes of the host and fungal strains; for example, the presence of *H. defensa* had no effect on the fungal sporulation or on the survival of pea aphids against *P. neoaphidis* [19], but, in contrast, it exhibited protection towards its host in our trials. In addition, *R. insecticola* potentially defends its host against fungal pathogens [15,21,54]. All these variations, including those noticed here, lead to the conclusion that the diverse functions performed by bacterial symbionts for their host aphids under various biotic and abiotic stresses have helped them to maintain their populations.

The possibility that bacterial secondary symbionts have the ability to act as a defensive wall for their hosts against natural enemies such as entomopathogenic fungi, parasitoids, and predators needs further consideration [23]. The mechanism behind *H. defensa* and *R. insecticola*’s resistance to fungal pathogens is still unknown, regardless of the symbiont’s genome availability [56]. It would be interesting to identify whether these two bacterial symbionts have developed these characteristics independently or whether there is some genetic aspect involved in their dissemination. There is also a probability that bacterial secondary symbionts boost the aphids’ defense against pathogens indirectly by galvanizing the immune system of the host and competing with pathogens for nutrition or proliferating essential nutrients accessible to the immune system [23]. In addition, the resistance conferred by bacterial symbiont genotypes against natural enemies may rely on abiotic components such as temperature [57], heat stress [58], and drought stress [59], and also host attributes such as age [60].

Our results broaden the significance of *H. defensa* and *R. insecticola* in wheat aphids, where they significantly reduced the mortality of wheat aphids caused by strains of entomopathogenic fungi *B. bassiana* (EABb04/01-Tip and Bb 1022) and *M. brunneum* (ART 2825 and BIPESCO 5). In addition to host aphid protection against parasitoids [21], *H. defensa* and *R. insecticola* also play a significant role in the host defense system by conferring resistance upon their hosts against fungal pathogens. This finding is another clue by which to understand the wide spread of these bacterial secondary symbionts among the insect communities. As these entomopathogenic fungi are being used as biocontrol tools against insect pests such as aphids [61], our work suggests that bacterial secondary symbiont-based protection against fungal pathogens may be a crucial consideration during the selection and development of biocontrol agents.

## 5. Conclusions

In general, researchers still need to develop a complete understanding of the role of bacterial secondary symbionts in insect host populations. Our results may assist in a better understanding of the interactions between BSS (*H. defensa* and *R. insecticola*) of *S. avenae* and EPF (*B. bassiana* and *M. brunneum*). It is concluded that these BSS provide their host aphids with increased resistance against the infection and toxicity caused by the application of EPF by lowering their mortality. Thus, it would be important to consider the presence of such bacterial secondary symbionts in the biological control of not only wheat aphids but also other insect pests of crops.

## Figures and Tables

**Figure 1 insects-13-01037-f001:**
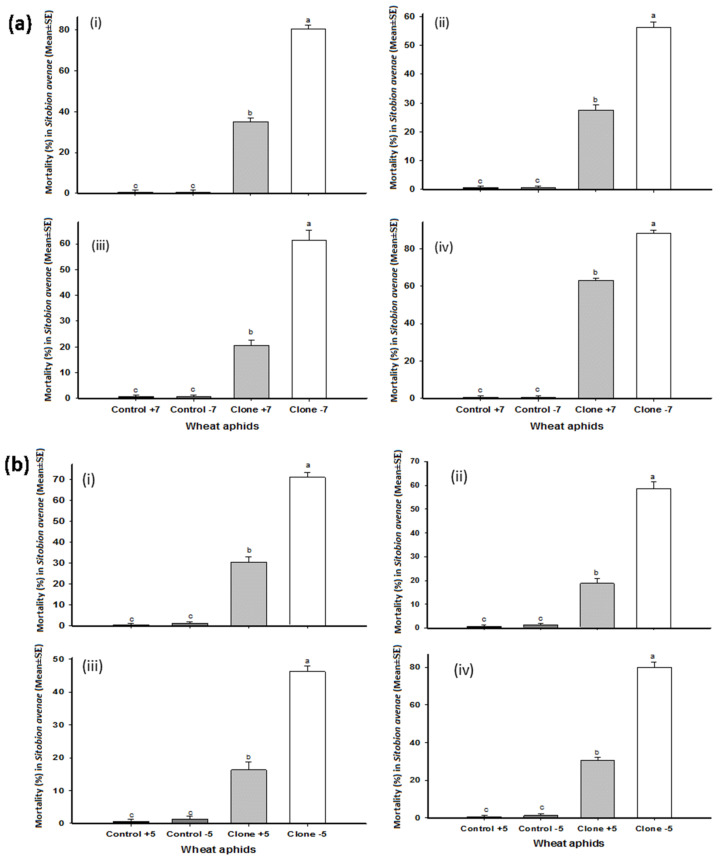
Total mortality in wheat aphid (*Sitobion avenae*) clones +7, −7 (**a**) and +5, −5 (**b**) after five days of application of *Beauveria bassiana* strains Bb1022 (i) and EABb04/01-Tip (ii) and *Metarhizium brunneum* strains ART 2825 (iii) and BIPESCO 5 (iv). The letters (a, b, c) are indicating the significance difference among the treatments.

**Figure 2 insects-13-01037-f002:**
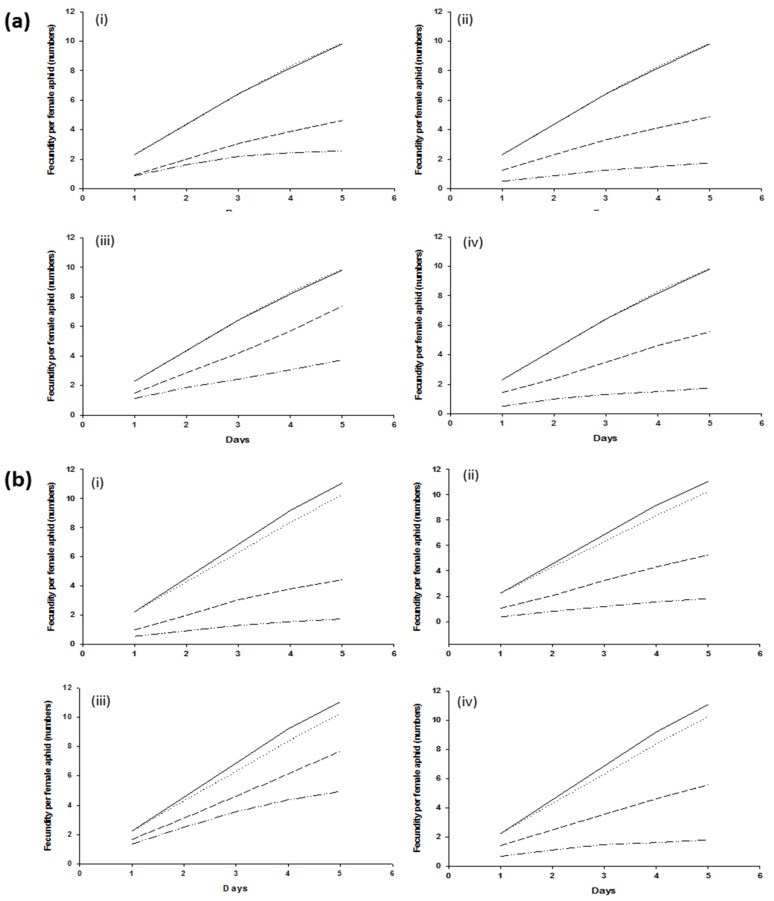
Number of offspring produced within 5 days following applications of entomopathogenic fungi *Beauveria bassiana* strains Bb1022 (i) and EABb04/01-Tip (ii) and *Metarhizium brunneum* strains ART 2825 (iii) and BIPESCO 5 (iv) to wheat aphid (*Sitobion avenae*) clones +5 and −5 (**a**) and clones +7 and −7 (**b**). (

) shows control with BSS, (
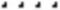
) shows control without BSS, (

) shows treatment with BSS, and (

) shows treatment without BSS.

**Figure 3 insects-13-01037-f003:**
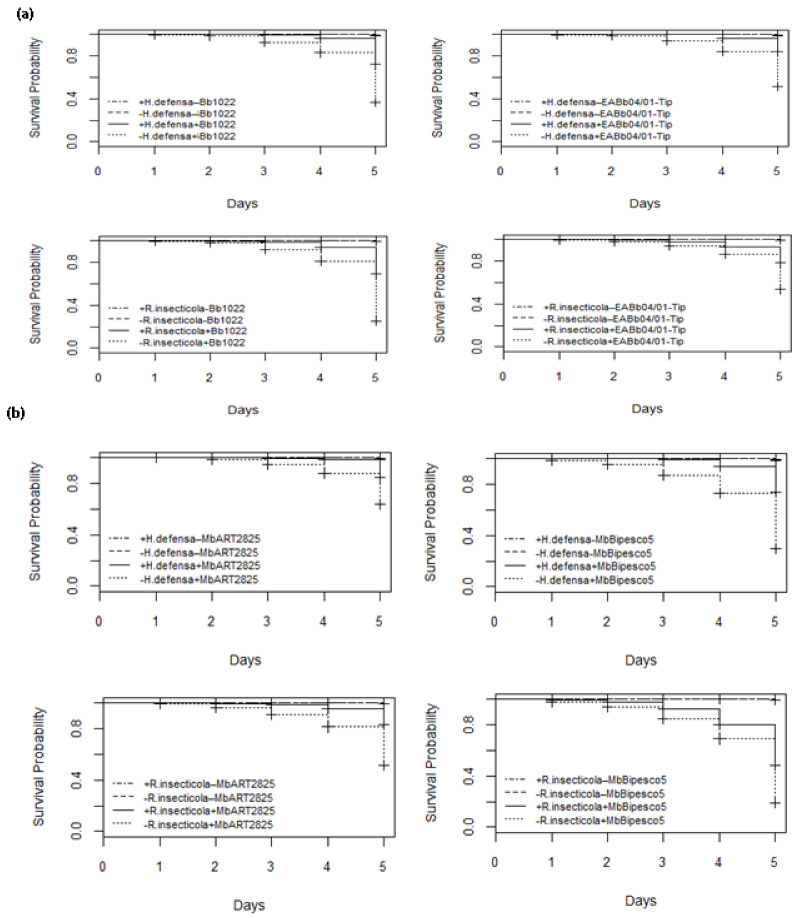
Survival curves showing infection period over 5 days post-application of entomopathogenic fungi *Beauveria bassiana* strains Bb1022 and EABb04/01-Tip (**a**) and *Metarhizium brunneum* strains ART 2825 and BIPESCO 5 (**b**) against wheat aphid (*Sitobion avenae*) clones +5, −5, +7, and −7.

**Table 1 insects-13-01037-t001:** Isolates of entomopathogenic fungi used in this study with geographic origin, original insect host.

Fungus Genotype	Strain	Geographic Origin	Insect Host	Reference
*Beauveria* *bassiana*	Bb1022	Canada	*Rhyacionia buoliana*	Vidal & Jaber, 2015 [47]
	EABb04/01-Tip	Spain	*Iraella luteipes*	Quesada-Moraga et al., 2009 [48]
*Metarrhizium* *brunneum*	BIPESCO 5/F52	Austria	*Cydia pomonella*	Eckard et al., 2014 [49]
	ART2825	Switzerland	*Agriotes obscurus*	Kölliker et al., 2011 [50]

**Table 2 insects-13-01037-t002:** Radial growth diameters (cm ± SE) of *Beauveria bassiana* and *Metarhizium brunneum* from dead wheat aphid (*Sitobion avenae*) clones with and without bacterial secondary symbionts.

	*Beauveria bassiana*				*Metarhizium brunneum*			
Days	Bb1022		EABb 04/01-Tip		ART 2825		BIPESCO 5	
	Clone +5	Clone −5	Clone +5	Clone −5	Clone +5	Clone −5	Clone +5	Clone −5
5	3.9 ± 0.2 a	3.6 ± 0.1 a	4.4 ± 0.2 a	4.3 ± 0.1 a	2.3 ± 0.2 b	3.5 ± 0.1 a	4.7 ± 0.2 a	5.3 ± 0.2 a
7	7.2 ± 0.1 a	7.1 ± 0.1 a	7.1 ± 0.2 a	7.5 ± 0.1 a	4.3 ± 0.3 b	5.5 ± 0.2 a	7.9 ± 0.4 b	8.9 ± 0.3 a
9	10.3 ± 0.2 a	10.3 ± 0.2 a	10.5 ± 0.2 a	10.6 ± 0.2 a	7.4 ± 0.4 a	7.9 ± 0.2 a	11.7 ± 0.4 a	11.8 ± 0.5 a
	Clone +7	Clone −7	Clone +7	Clone −7	Clone +7	Clone −7	Clone +7	Clone −7
5	3.3 ± 0.2 a	3.3 ± 0.1 a	4.2 ± 0.1 a	4.5 ± 0.1 a	2.9 ± 0.1 b	3.5 ± 0.3 a	4.1 ± 0.2 b	5.2 ± 0.2 a
7	6.7 ± 0.2 a	6.7 ± 0.2 a	7.2 ± 0.2 a	7.2 ± 0.1 a	4.5 ± 0.1 b	5.8 ± 0.4 a	7.6 ± 0.3 b	8.3 ± 0.3 a
9	9.3 ± 0.2 a	9.6 ± 0.2 a	9.8 ± 0.3 a	9.9 ± 0.2 a	6.9 ± 0.2 b	8.1 ± 0.4 a	10.9 ± 0.5 a	11.3 ± 0.4 a

Significant differences at the same times are indicated with different letters.

## Data Availability

Not applicable.

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
