# Peer review of "Influence of Bacterial Secondary Symbionts in Sitobion avenae on Its Survival Fitness against Entomopathogenic Fungi, Beauveria bassiana and Metarhizium brunneum"

_insects, 2022, doi:10.3390/insects13111037_

Round 1

Reviewer 1 Report

The manuscript describes the role of bacterial secondary symbionts of wheat aphids in defending the host insects from entomopathogenic fungal infection. The study was very interesting and well described.  Minor edits: L45: use singular form in keywords. L98: fast acting. L106: abbreviate entomopathogen fungi after first mention.
L119: The species Lecanicillium lecanii Zare & Gams has been re-classified as Akanthomyces lecanii (Zimm.) Spatafora, Kepler & B. Shrestha. Table 1: remove the middle row line inside the layout. L177: please describe the culture conditions. L189-190: without a solvent like Tween or Silwet, fungal conidia are difficult to mix with water. Using a magnetic stir for how long? It may not effectively separate conidia for uniform distribution in the suspension. L192: Were conidia checked for viability before test? L200: were the tests repeated or conducted once? L229: LSD test is sensitive to detect treatment differences. More conservative methods such as Tukey’s test or Duncan are recommended. L243: list F and df in statistical output in all Results. L275: In Fig. 1, label Y-axis as Mortality (%). Full spell species name at first mention; same in other Figures and Tables.

Author Response

Dear Reviewer,

thank you for your valuable suggestions.   You have highlighted good suggestions and improved MS by giving your valuable comments. We are thankful to you as a worthy reviewer for your time and suggestions. We accept all your indications as specified below.

Minor edits: L45: use singular form in keywords. L98: fast acting. L106: abbreviate entomopathogen fungi after first mention.

Response: The suggested changes have been incorporated in the Revised MS file.
L119: The species Lecanicillium lecanii Zare & Gams has been re-classified as Akanthomyces lecanii (Zimm.) Spatafora, Kepler & B. Shrestha. 

Response: Lecanicillium lecanii is replaced with reclassified name of that fugus in the Revised MS file.

Table 1: remove the middle row line inside the layout.

Response: The needful is done in the Revised MS file.

 L177: please describe the culture conditions. 

Response: The missing information is added in the Revised MS file.

L189-190: without a solvent like Tween or Silwet, fungal conidia are difficult to mix with water. Using a magnetic stir for how long? It may not effectively separate conidia for uniform distribution in the suspension.

Response: Actually, the sterile water was also containing tween 20 to make 0.03% water solution. The correction is made.

L192: Were conidia checked for viability before test?

Response: Actually, the spores were not checked for viability. We use the recommended concentrations from fresh fugal cultures which were being used for other experiments also.

 L200: were the tests repeated or conducted once?

Response: The treatments were replicated 16 times.

 L229: LSD test is sensitive to detect treatment differences. More conservative methods such as Tukey’s test or Duncan are recommended.

Response: LSD has more power compared to other post-hoc comparison methods (e.g., the honestly significant difference test, or Tukey test) because the α level for each comparison is not corrected for multiple comparisons (Williams and Abdi, Encyclopedia of Research Design.

Thousand Oaks, CA: Sage. 2010). The data sets and treatments number were not too large. This was performed for each EPF separately.

 L243: list F and df in statistical output in all Results.

Response: The suggested changes are done.

 L275: In Fig. 1, label Y-axis as Mortality (%). Full spell species name at first mention; same in other Figures and Tables.

Response: The suggested changes have been added.

Reviewer 2 Report

The manuscript entitled "Influence of bacterial secondary symbionts in Sitobion avenae on its survival fitness against entomopathogenic fungi, Beauveria bassiana and Metarhizium brunneum" is appropriate for the journal. This research provides new knowledge on the interaction of bacteria as secondary symbionts of Sitobion avenae and fungal pathogenesis, a phenomenon that should be considered for the selection of microbial control agents. The experiments seem appropriate, however some sections require attention to improve their presentation. Specific comments will be mentioned in the manuscript.

Author Response

The reviewer has raised good points and improved our MS by giving his valuable comments and suggestions. We are thankful to the worthy reviewer for his time and suggestions.

Below is specified the changes made.

 L177: PDA product trademark? complete names and taxonomic classification

Response: The trademark is provided in revised MS file.

 L181: complete names and taxonomic classification of the highlighted names?

Response: The suggested changes have been made.

 L192: exp

Response: The dose is corrected.

L196: indicate distance and infective units per mm2

Response: distance is given and concentration is corrected.

L202: Tween TM 20??

Response: It is corrected.

L221: Refer final volume and trademarks of antibiotics?

Response: It is corrected.

L281: The study refers to radial growth, or the invasiveness of the strains?

Response: It is corrected.

L305: Retitle the table 2? move as table footer

Response: The title is retitled.

L306: move this line as table footer!

Response: The suggested lines are moved as footer.

L306: Your conclusion must be specific, in relation to your most relevant results!

Response: The conclusion is rewritten as per suggestion of the worthy reviewer.